# The Stress Response of the Holothurian Central Nervous System: A Transcriptomic Analysis

**DOI:** 10.3390/ijms232113393

**Published:** 2022-11-02

**Authors:** Sebastián Cruz-González, Eduardo Quesada-Díaz, Yamil Miranda-Negrón, Raúl García-Rosario, Humberto Ortiz-Zuazaga, José E. García-Arrarás

**Affiliations:** 1Department of Biology, College of Natural Sciences, University of Puerto Rico, Río Piedras, San Juan, PR 00925, USA; 2Department of Computer Science, College of Natural Sciences, University of Puerto Rico, Río Piedras, San Juan, PR 00925, USA

**Keywords:** echinoderm, heat shock proteins, ubiquitin, regeneration, RNA-seq, spinal cord injury

## Abstract

Injury to the central nervous system (CNS) results in permanent damage and lack of function in most vertebrate animals, due to their limited regenerative capacities. In contrast, echinoderms can fully regenerate their radial nerve cord (RNC) following transection, with little to no scarring. Investigators have associated the regenerative capacity of some organisms to the stress response and inflammation produced by the injury. Here, we explore the gene activation profile of the stressed holothurian CNS. To do this, we performed RNA sequencing on isolated RNC explants submitted to the stress of transection and enzyme dissection and compared them with explants kept in culture for 3 days following dissection. We describe stress-associated genes, including members of heat-shock families, ubiquitin-related pathways, transposons, and apoptosis that were differentially expressed. Surprisingly, the stress response does not induce apoptosis in this system. Other genes associated with stress in other animal models, such as hero proteins and those associated with the integrated stress response, were not found to be differentially expressed either. Our results provide a new viewpoint on the stress response in the nervous system of an organism with amazing regenerative capacities. This is the first step in deciphering the molecular processes that allow echinoderms to undergo fully functional CNS regeneration, and also provides a comparative view of the stress response in other organisms.

## 1. Introduction

In higher vertebrates such as humans, severe trauma to their central nervous system (CNS) can lead to life-long paralysis, whereas less complex organisms are capable of extensive CNS regeneration [1]. The reason behind such significant differences in regenerative potential remains unclear. However, recent studies have pinpointed the initial stress response to CNS trauma as a trigger of subsequent events that may limit overall tissue regeneration [2]. Paradoxically, the very mechanisms of the CNS that respond to damage in higher vertebrates are what lead to its regenerative incompetence. At the cellular and transcriptomic level, past research has often divided the body’s response to spinal cord injury (SCI), a common CNS lesion, into an acute phase (~24 h), subacute phase (2–7 days), and chronic phase (weeks, months) [3,4,5,6,7]. The acute phase corresponds to the direct effect of injury on the tissue, whereas both the subacute and chronic phases are mediated by the immune system or other events that originate from the acute response. The early response to trauma is significant as it is essential for wound healing and clearing of debris at the injury site; however, if this process becomes prolonged or excessive, it can lead to secondary damage such as glial scarring. This scarring creates an unfavorable environment for tissue renewal [8]. Experiments in non-mammalian species, particularly fishes and amphibians, have shown some peculiarities that may account for the increased regenerative potential of these species [9,10,11,12,13]. Thus, investigators have probed the gene expression profiles of species that can regenerate their spinal cord following transection, as a source of insight (and future therapies) to understand the limits of CNS regeneration in mammals.

Organisms in the phylum Echinodermata hold a unique evolutionary position, that places them at a basal branch in vertebrate evolution. Their nervous system has been described by several investigators [14,15]. In brief, their CNS is composed of five radial nerve cords (RNCs) that come together in an anterior nerve ring arranged around a central axis. The radial nerves are ganglionated, containing neuronal and glial cell bodies surrounding a central neuropil area [16]. However deceptively simple this arrangement may seem, various subpopulations of cells give these tissues amazing functional complexity, even in the absence of cephalization or specialized nervous structures. Furthermore, these organisms are capable of extensive CNS regeneration [17,18]. Their regenerative capabilities, relatively simple anatomy, and close relation to vertebrates make these organisms ideal models for biomedical studies on CNS regeneration. A model based on these organisms would allow researchers to study the mechanisms by which these animals minimize neurodegeneration and trigger a regenerative process following injury.

The brown rock sea cucumber, *Holothuria glaberrima*, has been a particular focus of past studies. These studies have investigated the morphological characterization of the regenerating RNC structures, as well as described the cellular events that underlie the regeneration process [17]. One of the main findings arising from these studies is the essential role that radial glia-like cells play in mediating CNS regeneration by giving rise to neurons and other cells through dedifferentiation and proliferation [18,19]. Although these studies described the cellular events that unfold during RNC regeneration and the distinct populations of cells present during various regenerative stages in detail, there is limited information on the molecular events that occur during regeneration. Mashanov and colleagues provided an overview of the transcriptomic changes during RNC regeneration at various stages in *H. glaberrima* (specifically at 2, 12, and 20 days post-injury), and identified the groups of genes that were found to be differentially expressed [1]. However, the study suffered from a possible confounding effect, as the tissues for transcriptomic analysis were not exclusively RNCs. Due to the anatomical structure of the RNCs in holothurians, surgical extraction of an isolated RNC is practically impossible and the RNC dissection includes mRNA from the surrounding connective tissues, body wall muscles, and the water vascular canal, among others. This can lead to misidentification of the differentially expressed genes found at each stage, as the genes may be differentially expressed in some other tissue, but not the actual RNC.

We have now developed a protocol to isolate *H. glaberima* RNCs by collagenase treatment [20]. This technique allows almost complete isolation of RNCs from surrounding tissues, thus producing a source of CNS tissue with little or no contamination from adjacent tissues. The protocol, nonetheless, is somewhat abrasive, and produces an acute stress response on the isolated RNCs akin to the initial response to trauma experienced by the RNC after injury. Therefore, we have used this protocol to characterize the molecular basis of this stress response to understand the initial response to trauma of animals that exhibit extraordinary capabilities for CNS regeneration.

## 2. Results and Discussion

### 2.1. Stress Response Genes of the Holothurian Central Nervous System

To identify genes of interest involved in the holothurian CNS stress response, we compared the gene expression of RNCs following their dissection and extraction from the body wall by a 24-h collagenase treatment with that of RNC explants kept in culture for 3 days following collagenase dissection [20]. The rationale behind this experiment was that the transection of the RNCs from anterior and posterior regions and from the peripheral nerves, together with the 24-h collagenase treatment would induce a stress response in the RNC cells that would be ameliorated as the tissues were kept in tissue culture for the next 3 days. Differential gene expression analysis was performed using the *DESeq2* R package (version 1.27.12; Michael I. Love; Boston, Massachusetts, USA) [21]. We identified 422 genes with a significant increase in expression (log_2_ Fold Change ≥ 2 and FDR-adjusted *p*-value < 0.05) at 24 h post-collagenase treatment, relative to the level of expression after 3 days in tissue culture. Conversely, we identified 454 genes that showed a significant decrease in expression (log_2_ Fold Change ≤ −2 and FDR-adjusted *p*-value < 0.05) during the same time frame (Figure 1).

Differentially expressed genes were then used to create two fasta files for annotation using a Python script provided by the developers of the *Corset* software (version 1.06; Davidson and Oshlack; Melbourne, Australia) [22]. The two fasta files were then annotated using the *BLAST* command line tool [23]; the Uniprot peptides database was used as our reference database [24]. The 30 genes with the most significant differential expression, annotated using the Uniprot database, are presented in Table 1.

### 2.2. Functional Groups of Genes Involved in the Stress Response of the Holothurian Central Nervous System

Functional groups of genes were identified according to Gene Ontology (GO) Consortium terminology. For upregulated genes, enriched terms in molecular functions (MF) included unfolded protein binding, peptidase activity, and transmembrane receptor activity. Biological process (BP)-enriched terms ranged from cellular responses to stimuli, and from proteolysis, to cell adhesion. Finally, enriched terms in cellular components (CC) included membrane parts and integral components of the membrane (transmembrane protein) (Figure 2A).

Conversely, enriched terms in MF for downregulated genes included RNA polymerase activity, oxidoreductase activity, and transferase activity. BP-enriched terms encompassed RNA processing, ncRNA processing, and ribosome biogenesis. Lastly, CC-enriched terms ranged from mitochondrial envelope to intracellular organelle parts, and membrane protein complex (Figure 2B).

These initial analyses provided a broad overview of the most prevalent processes during the CNS stress response in *H. glaberrima*. The enriched GO terms served to identify genes that were associated with certain cellular processes of the stressed tissues by either increasing or decreasing their expression. It was interesting to compare these results with those obtained after holothurian RNC transection in vivo [1]. In fact, there was little, if any, similarities between the processes defined by their gene expression profiles. In vivo, most processes 2 days after transection corresponded to processes associated with the extracellular matrix or muscle systems. The reason behind this is that, in vivo, the RNC is embedded within the body wall and is practically impossible to completely isolate the tissue. Thus, the in vivo results most likely represented the ongoing changes in the body wall tissues with little, if any, contribution from the RNC. In this respect, our present results, showing biological terms associated with the response to various stressors, correspond to a factual view of what is taking place within the RNC neuronal and glial cells after transection, as little to no other tissue contamination is present in our in vitro explants.

The enriched terms, however, give only a general idea of what is occurring at a molecular level during the stress response. To provide a more accurate description of the factors involved, we surveyed the RNC transcriptome database for homologs that matched classical stress actors in ubiquitination, apoptosis, and unfolded protein responses, along with other recently described families of stress-associated proteins, such as the stress-protective factors *Hero* [25].

### 2.3. Heat Shock Proteins

The heat shock protein (Hsp) family has been associated with not only heat-induced stress, but also stress originating from other environmental factors. This protein family is divided into five groups according to size and sequence similarities: the 60-, 70-, 90-, 100-kDa Hsps and the small size Hsps (sHsps). Therefore, we decided to look for sequences that matched these and other members of the Hsp family, as well as additional proteins commonly associated with Hsp family members. As expected, potential homolog sequences were highly represented in our transcriptome and showed significant upregulation after 24 h of collagenase treatment.

We identified one putative *Hsp20* sequence in the RNC transcriptome database. The sequence was similar to that of the *Hsp20* sequence of the purple sea urchin *Strongylocentrotus purpuratus*. This sequence showed upregulated expression after collagenase treatment.

We next focused on *Hsp90* homologs. We found one sequence similar to the *Hsp90* protein (Figure 3) that aligned with a cytosolic *Hsp90* protein from *A. japonicus*. We were unable to detect isoforms of the *Hsp90* proteins that were found in the mitochondria or endoplasmic reticulum [26]. The aforementioned contig was found to be significantly upregulated 24 h after RNC transection and collagenase treatment.

In our probe, we also found a contig with very high similarity to the chaperone *DNAJA1B* (also known as *Hsp40*) protein of *A. japonicus*. This contig was complete and possessed significant upregulation of expression after RNC transection and enzymatic dissection. A second partial sequence with significant homology to *DNAJ*, and distinct from the first contig, was also found to be upregulated after collagenase treatment.

Rounding off our search for members of the classical Hsps, we probed the transcriptome for *Hsp70* homologs. For this class of *Hsps*, we found five distinct contigs (28% homology, 19% identity amongst themselves), all similar to *Hsp70* proteins from various organisms. An alignment of the conserved *Hsp70* functional domain of the five sequences is shown in Figure 4. Two of these contigs contained a complete sequence, meaning they possessed an initial methionine and a stop codon, whereas the remaining sequences were partial. All the aforementioned contigs, whether complete or partial sequences, possessed significant upregulation of expression at 24 h post-collagenase treatment. The identification of various distinct *Hsp70* family members in the RNC transcriptome of the *H. glaberrima* was consistent with the literature on this family of proteins, where multiple isoforms have been identified for *D. melanogaster*, *C. elegans*, *Ciona intestinalis*, and *H. sapiens* [27]. In fact, several Hsps have been characterized from different sea cucumber species. Among these are Hsp 90, Hsp 70, Hsp 60, and members of the small Hsp family [28,29,30,31,32]. These genes have been reported to be expressed by cells in the intestine, coelomocytes, and other tissues. Similarly, they are differentially expressed during stress, mainly due to temperature, salinity, hypoxia, or immune activation. Still, few reports associate these genes with regenerative processes, and none have shown their expression in the nervous components of echinoderms. Some of the relevant findings were reports demonstrating Hsp72 upregulation in the regenerating arms of crinoids, another echinoderm class [33], and changes observed in various Hsps during wound healing in the sea cucumber *H. tubulosa* [30].

We also searched for lesser-known chaperones or putative heat shock proteins. In this probe, we identified a homolog for the sacsin chaperone in the RNC transcriptome. The contig was similar to the *DNAJC29* sacsin from *H. sapiens* and *M. musculus*. This protein is a regulator of the *Hsp70* machinery, and therefore, an important regulator of stress. The interplay between *Hsp70* and various co-chaperones has been shown to protect from neurotoxicity by promoting ubiquitination and degradation of misfolded proteins to prevent aggregation in models of neurodegenerative diseases [34,35].

Although many chaperone proteins were upregulated following RNC dissection, we identified some exceptions to these findings. The first example to catch our attention was a homolog for *CALR*, which encodes the calreticulin protein. This expression pattern was paradoxical because calreticulin is another important quality control mediator, specifically in the endoplasmic reticulum. This is made all the more striking when one considers that this protein is necessary for post-stress survival in *Caenorhabditis elegans* in vivo [36]. A possible explanation may be that calreticulin is necessary for a long-term stress response and that its levels remained elevated in the explant in the 3 days after dissection by collagenase treatment.

Other downregulated factors that caught our attention were various mitochondrial chaperones. Among these were homologs for *BCS1* and *TIMM10*. Both contigs were significantly downregulated after transection and 24 h post-collagenase dissection. These data, along with the data shown in Figure 2B, suggested a depletion in mitochondrial activity of these cells following stress. Davis et al. [37] and Hayakawa et al. [38] have described a phenomenon by which neurons transfer damaged mitochondria to surrounding glial cells for lysosome degradation. This form of cell-to-cell signaling could potentially explain the depletion of mitochondrial transcriptomic activity at 24 h post-collagenase treatment. As this transfer of mitochondria is two-sided, meaning that neurons may receive new mitochondria from glial cells, this could also explain the observed cell viability and low apoptotic index.

The protective effect of individual Hsps may vary depending on the stress factor and cell type. Nevertheless, there were several reports where the expression of Hsps was associated with nervous system development and/or its protection during disease, injury, or other stressors [39,40,41,42]. Some of the proposed nervous system roles of Hsps include: suppressing inflammation, increasing survival, modulating differentiation and/or neurogenesis, and regenerative processes in holothurian CNS neurons.

In summary, with the exception of calreticulin and mitochondria-specific chaperone proteins, Hsps were mostly upregulated following CNS transection and enzymatic dissection, suggesting these types of protein factors play an essential role in the stress response of the holothurian CNS.

### 2.4. Ubiquitin

Chaperones are known to promote the ubiquitination and degradation of misfolded proteins. Thus, it is not surprising that we found that many of the proteins associated with the ubiquitin proteasome system were also upregulated following RNC enzymatic dissection. The ubiquitin pathway is inextricably associated with stress responses at the molecular level in higher eukaryotes. The pathway plays an important role in damaged protein disposal during stress responses to prevent harmful protein aggregation and to help restore homeostasis in the long-term [43]. Previous work from our group and others have provided insights on the role that this pathway may play in stress responses of echinoderms [44,45,46]. Using these previous studies to confirm what we observed in our NGS data, we decided to survey various components of this pathway, and other ubiquitin-associated factors, in our RNC transcriptome. We found various matches for ubiquitin and polyubiquitin that were upregulated 24 h post-collagenase treatment, as well as various protein factors involved in the ligation of ubiquitin to defective protein products. Among these were RBBP6, RNF31, UBE3C-E3, TRIM2, and β-TrCP. We also identified the upregulation of various protein factors known to be associated with the ubiquitin proteolysis pathway at 24 h post-collagenase treatment. These factors included DYRK2 and ZFAND2B. Conversely, we identified inhibitors of ubiquitination among the downregulated genes at 24 h post-collagenase treatment. Such genes included NOP53, UBLCP1, and BABAM1. From this list of downregulated genes, we also identified various factors involved in protein ubiquitination, such as UBFD1, UBE2C, STUB1, CUL5, and UBE2U. Finally, we identified a transcript with similarities to STUB1 with a significant downregulation in expression. This particular transcript encoded an E3 ubiquitin ligase, which was the only transcript of an E3 ligase to be downregulated at 24 h post-collagenase treatment.

In these analyses, we identified 13 transcripts with similarity to ubiquitin that were upregulated 24 h post-collagenase treatment. To elucidate the uniqueness of each of these transcripts, we aligned all 13 translated transcripts using the Geneious software. This alignment showed that the transcripts formed three subgroups. The first subdivision contained a single sequence including an initial methionine and stop codon. The second subdivision contained two sequences that shared 50% similarity with each other, but less than 50% similarity with the rest of the surveyed transcripts; only one of these sequences contained an initial methionine, but did not have a stop codon. Finally, the third subdivision had nine sequences that were 43% similar and 39% identical amongst themselves. From these nine sequences, we selected the most representative sequence based on its length and sequence similarity shared with the sequences from the other two sequence subdivisions. The remaining eight sequences might represent distinct alleles or sequencing errors, so we decided to focus on only one sequence from this group for the present study. In summary, we identified at least four distinct transcripts that encoded different ubiquitin proteins in *H. glaberrima* (Figure 5 and Appendix A). The possibility that other ubiquitin isoforms are present is likely, as our analysis was highly conservative trying to identify only the most distinct sequences.

As ubiquitination for proteasomal degradation is a common molecular hallmark of stress in eukaryotes, we expected many components of this pathway to be significantly upregulated at 24 h post-collagenase treatment. Consistent with our expectations, many upregulated factors included genes that code for the actual ubiquitin and polyubiquitin molecules, as well as various proteins involved in ubiquitin ligation. Our data also uncovered a complex network of these proteins, where some ubiquitin ligases were upregulated, but others experienced significant downregulation during the CNS stress response. We hypothesized that this molecular process within the cells of the holothurian CNS involved the ubiquitination of only specific groups of proteins.

The identification of at least four distinct ubiquitin transcripts further supported the idea of intricate regulation of proteasomal degradation, as the four transcripts shared a certain degree of similarity (owed to their ubiquitin domain), but with interesting differences in the remaining sequences. Such differences in sequence composition are a classic biological mechanism for specificity, particularly in binding specificity and affinity. Another interesting possibility is that some of these transcripts may be pseudogenes. One such pseudogene (*UBBP4*) with ubiquitin domains that is actively transcribed was recently described by Dubois et al. [47]. Though we identified at least four transcripts, we do not rule out the possibility of more transcripts being present in our RNC transcriptome, whether they are protein-coding variants that allow further specificity or pseudogene transcripts with yet to be described functions.

Following up on the identification of various upregulated ubiquitin transcripts at 24 h post-collagenase treatment, we expected to find that E3 ubiquitin ligases, the final effectors of the ubiquitin tagging modification, would follow the same pattern of transcript abundance. This was true, to a certain extent, given our identification of three E3 ligases that were upregulated at 24 h post-collagenase treatment: RNF31, UBE3C-E3, and TRIM2.

An exception to the expression patterns observed for most of the E3 ubiquitin ligases was that of STUB1. Though all the other E3 ligases were upregulated at 24 h post-collagenase treatment, STUB1 was downregulated at this timepoint. A possible explanation for this is that STUB1 is a versatile protein that plays various roles in the stress response. Among these functions is E4 ligase-like activity, wherein STUB1 regulates other E3 ligases in the stepwise process of ubiquitination and proteasome degradation [48]. Therefore, the time of action for STUB1 in the RNC stress response may be later than that of the other E3 ligases due to functional plasticity. It is also worth pointing out that STUB1 has also been found to mediate ubiquitination of Hsp70 and Hsp90 chaperones for proteasomal degradation [49]. The pattern of expression found for STUB1 is then consistent with the upregulation of these chaperones in the RNC transcriptome at 24 h post-collagenase treatment.

### 2.5. Apoptosis

Programmed cell death or apoptosis is closely related to cellular stress responses. Therefore, it was important to determine the expression of genes associated with apoptosis in our transcriptomic data. Several apoptotic related genes, namely *TRAF3*, *AP-1*, *SQSTM-1*, *BAK1*, *CCAR1*, and *LAMTOR-5,* were found to be differentially expressed (Table 2). However, several other apoptosis-related genes, including *BIRC5* (*survivin*), a member of the *IAP* family, *BCL-2*, and *p53,* showed no significant differential expression. Similarly, a search for caspases (as these proteins are some of the main effectors of cell death) in our transcriptome identified homologs for caspases 1–3 and 6–8, but none were differentially expressed. These data appear paradoxical at first glance, with some apoptotic genes showing differential expression while others show no change in expression. In brief, our findings provided no clear clue as to whether apoptosis was being induced by the stress response or not. To settle the issue, we performed a TUNEL analysis to quantitate apoptosis at the cellular level. In both collagenase-treated tissues (24 h) and explants left for 3 days in culture, we found a small number of apoptotic cells in RNC explants. The apoptotic index was quite low (<1%) (Figure 6), which could be explained due to the action of chaperones [50]. Overall, our data suggest that though some genes in the apoptosis pathway were differentially expressed, apoptosis was not actively occurring during the stress response. This observation was surprising, as stress is commonly known to mediate cell death through careful orchestration by the endoplasmic reticulum and the unfolded protein response (UPR). Studies from mammals indicate that modulation of apoptosis exerted by the UPR may be responsible for this paradoxical outcome [51]. In fact, we found XBP1, a factor involved in the UPR, was upregulated at 24 h post-collagenase treatment. A contig homologous to *XBP1* was found to be significantly upregulated at 24 h post-collagenase treatment. *XBP-1* encodes a transcription factor essential for the UPR in the ER. Specifically, the activated *XBP-1* protein regulates a range of transcriptional targets, including *GATA3*, *MYC*, and *FOS* [52]. In this process, the *XBP1* transcription factor can act as a “switch” between apoptosis and survivability by regulating its downstream effectors. In various Alzheimer’s Disease models, the active form of *XBP-1* has shown neuroprotective activity by way of preventing amyloid-beta neurotoxicity after ER stress [53], a fact that highlights the importance of this factor in the prevention of stress-mediated neurodegeneration. To our knowledge, this is the first piece of evidence supporting the presence of this apoptosis repression pathway and associated physiological responses in echinoderms. The broader implications of these data will be of great interest for future work, given that we have shown that RNCs that are transected and treated with collagenase for 24 h display many of the traditional hallmarks of stress (widespread ubiquitination, Hsp activation), but avoid the pitfall of apoptosis. Future experiments in our laboratory will provide data for appropriate comparisons of this apoptosis repression response in *H. glaberrima* to that of other mammals where it has been comprehensively described.

### 2.6. Other Stress-Related Genes

#### 2.6.1. Retrotransposons

Retrotransposons have been implicated in the response of cells to stress [54]. In previous experiments, Mashanov and colleagues identified a series of long terminal repeat (LTR) retrotransposons in the radial nerve cords of *H. glaberrima* [55]. Since the involvement of these LTR retrotransposons in neural regeneration was a key finding of their work, we decided to survey the presence of the 36 sequences in our transcriptome database. We found all 36 LTR sequences. However, only two of the sequences were differentially expressed relative to the *Day3* group: *Gypsy-19* and *Gypsy-1*, with the former being upregulated and the latter downregulated. In vivo, it was shown that the expression of most retrotransposons remains high for 1–3 weeks, thus it is possible that in our in vitro explants, the expression remains high and therefore shows no difference from the recently injured/enzymatically dissociated RNC. In this respect it is interesting that *Gypsy-19*, in vivo, is one of the few transposons that shows and acute response increasing almost 4-fold 2 days after injury, but then going back to uninjured levels by the following week.

#### 2.6.2. Hero Proteins

Recently, Tsuboyama et al. [25] described the *Hero* protein family, a group of heat-soluble molecules found in non-extremophilic organisms. These proteins function as shields for heat-susceptible proteins, protecting them under stressful conditions such as heat-shock and desiccation, and thus counteracting protein denaturation. Moreover, this family of proteins plays a role in preventing neurodegeneration; in *Drosophila melanogaster* neurodegeneration models, these proteins block the pathological aggregation of proteins observed in various neurodegenerative diseases. Because this family of proteins shows promise for studies of neural diseases, we decided to search for homologs in the transcriptome of the isolated RNCs.

To perform this search, we used the BLAST software, querying the RNC transcriptome with the Uniprot database (which itself holds the *Hero* protein sequences). As the *Hero* protein family is not necessarily defined by a specific primary structure or motif, it is difficult to identify homologs in *H. glaberrima* for this novel family of proteins. Nevertheless, we have identified three homologs in the radial nerve cord transcriptome. Two of the homologs (*Hero45*, *Hero11*) did not experience significant differential expression at 24 h post-collagenase treatment, and the homolog that did experience significant differential expression (*Hero7*) was downregulated post-collagenase treatment. This observation contrasts with our expectations for the role these proteins play in neurodegeneration.

#### 2.6.3. Integrated Stress Response

The integrated stress response (ISR) pathway has been described as an important mechanism used by eukaryotic cells to mitigate extrinsic (e.g., nutrient deprivation) and intrinsic (e.g., ER stress) cell stressors [56]. Pathway initiation is regulated by phosphorylation of eukaryotic translation initiation factor 2 alpha (eIF2α) via PERK, PKR, HRI, and GCN2 kinases [57]. Once eIF2α is phosphorylated, global protein synthesis is reduced, and survival and recovery genes, such as activating transcription factor 4 (ATF4), are selectively translated [56]. Given the physiological stress exerted on *H. glaberrima* RNC, we evaluated ISR pathway involvement by assessing 10 main ISR factors (Table 3). All factors, except C/EBP homologous protein (CHOP), were found in our transcriptome database. To our surprise, none of the listed factors showed differential gene expression compared with the RNC explant cultured 3 days post-collagenase treatment. These results are consistent with our previous data. It is well-known that apoptotic factors p53, BCL-2, and caspase-1 are effectors of the ISR pathway, none of which showed significant differential expression. Therefore, the accumulation of these results suggests no active ISR signaling pathway in collagenase-treated RNC.

#### 2.6.4. Transcription Factors

We identified various differentially expressed contig sequences homologous to an array of transcription factors (Table 4). Among these were *ATF2*, a transcription factor that dimerizes with *Jun* [58] to regulate cellular stress responses, including the DNA damage response [59]. *RXRA* was another upregulated factor identified in our transcriptome database. This factor is primarily a retinoid X receptor [60], but also shuffles between the nucleus, cytoplasm, and mitochondria. *RXRA* has been shown to translocate to dimerize with *NUR77*, after which it is translocated to the mitochondria, and *NUR77* interacts with *BCL-2* to induce apoptosis [61]. None of these transcription factors were reported to be differentially expressed after transection in vivo [1], probably due to what was explained earlier with other components (muscle, ECM) within in vivo samples biasing measures of gene expression.

## 3. Materials and Methods

### 3.1. Sample Collection

Adult individuals of the sea cucumber *Holothuria glaberrima* were collected from the north coast of Puerto Rico. Specimens were then stored in sea water aquaria at room temperature for 24 h. Samples from *H. glaberrima* specimens were collected by excising radial nerve cords (RNCs) along with surrounding tissues. The RNCs are anatomically located close to various tissues (such as connective structures, segments of the body wall, and segments of the water-vascular canal), making perfect isolation via surgery practically impossible. Given this limitation, we used a recently described protocol to isolate the RNC [20]. In brief, the excised tissue complexes were placed in collagenase for 24 h. This treatment isolates RNCs by degrading the collagen composition of the surrounding tissue. The surgical transection of the RNC from other CNS structures and peripheral nerves incites a stress response from the nervous tissue, allowing characterization of the central nervous system (CNS) stress response in our model organism. Therefore, comparisons were made between RNCs that had been collagenase-treated for 24 h and RNCs that were cultured as explants for 3 days following the collagenase treatment.

### 3.2. RNA Extraction

Total RNA extraction was performed using phenol and chloroform standard reagents, as well as the RNeasy mini kit (Qiagen, Hilden, Germany). The concentration and purity of the RNA was determined using the NanoDrop-1000 Spectrophotometer (Nano Drop Technologies, Rockland, DE, USA). Various RNC fragments from 1–2 specimens were pooled together to gather RNA and were treated as individual samples. Thus, 5 biological replicates were prepared: three corresponding to the RNCs treated with collagenase for 24 h (0 day group) and two corresponding to the RNCs cultured for 3 days (3 day group) following collagenase treatment.

### 3.3. RNA Sequencing

Prepared RNA samples were sent to the Sequencing and Genomics Facility of the University of Puerto Rico in Rio Piedras for library preparation and RNA sequencing, as previously shown [62]. Libraries were prepared using the TruSeq RNA Library Prep Kit (Illumina, San Diego, CA, USA). Subsequently, paired-end RNA sequencing was performed (Illumina NextSeq 500, Illumina). Raw sequencing reads from the Illumina platform were deposited at blastkit.hpcf.upr.edu/hglaberrima-v1 (accessed on 2 July 2020).

### 3.4. De Novo Assembly Pipeline

The *elvers* software was used for the majority of the RNAseq analyses performed on the RNC data. The standard workflow (*Eel Pond Protocol*; version 1.0; Brown, Scott, and Sheneman; Davis, CA, USA) defined by the developers was followed. This includes quality filtering, quality assessment, k-mer trimming and digital normalization, de novo assembly, read quantification, and differential expression analyses. All transcriptome processing was performed with the *elvers* (version 0.1; Scott; Davis, CA, USA) modules, except for transcriptome annotation and hierarchical clustering of assembled contigs, which were performed separately. Additional differential expression analyses were also performed without the use of the *elvers* differential expression analysis module.

Reads were quality trimmed using the *Trimmomatic* software (version 0.30; Bolger, Lohse, and Usadel; Jülich, Germany). Using Illumina adapters, leading and trailing bases with quality < 2 were removed from the sequencing data, as well as reads shorter than 25 base pairs.

K-mer trimming and digital normalization for the quality-trimmed reads were performed to reduce the computational burden of transcriptome assembly. A Python script provided by the *khmer* module from the *elvers* software was utilized, with a *k* of 20. If a particular k-mer was identified more than three times in the data, it was removed, thus decreasing the amount of reads for assembly without forsaking the quality of the assembled transcriptome.

The *Trinity* software (version 2.8.6; Grabherr; Boston, MA, USA) was utilized for transcriptome assembly. Quality and k-mer trimmed paired-end reads were provided as input, and default parameters were used to generate contiguous sequences.

Quality-trimmed reads from the five sequenced libraries were aligned to the assembled contigs using the *Salmon* software (version 1.9.0; Patro; Stony Brook, NY, USA) to quantify expression. The *–dumpEq* parameter was specified to obtain equivalence classes. These classes were necessary for contig hierarchical clustering via the use of the *Corset* software. This step aimed to solve the redundancy issue that is common to de novo transcriptome assemblers such as *Trinity*. The *Corset* software hierarchically clustered contigs based on shared reads and expression, summarizing read counts into clusters that could then be interpreted as “genes”. *Corset* was utilized with default parameters.

To perform the differential expression gene analysis, the *DESeq2* R package was utilized. A comparison was made taking the 0 day group as the experimental group and the 3 day group as a control group. To define a gene as differentially expressed, we established a false discovery rate (FDR) threshold of <0.05.

### 3.5. Transcriptome Annotation

The assembled RNC transcriptome was annotated using the BLAST command line software. The Uniprot database was utilized as a reference for identifying sequences of interest. Homology for our contig sequences was defined via the use of two parameters: E-value < 1 × 10^−5^ and 50% identity between the matched sequence in the Uniprot database and our contig sequence. Additional annotation for ubiquitin and Hsp family proteins was also performed using the NCBI non-redundant (nr) database, with the established parameters for defining matches.

### 3.6. Gene Set Enrichment Analysis and KEGG Pathway Enrichment

The *DAVID* (Database for Annotation, Visualization, and Integrated Discovery) software was used to perform functional annotation, identifying groups of genes and molecular pathways that were significantly enriched during RNC stress response [63]. Particularly, the *RDAVIDWebService* (version 1.0.0; Fresno; Córdoba, Argentina) R package was used to access the data available from *DAVID* (version 6.8; Sherman; Frederick, MD, USA) databases within the R interface. The obtained IDs from the Uniprot BLAST were used as the gene background for enrichment analyses, and the following annotation categories were specified: “GOTERM_BP_ALL”, “GOTERM_MF_ALL”, and “GOTERM_CC_ALL”. The obtained GO terms were filtered using the Revigo software (version 1.0.0; Supek; Zagreb, Croatia) to eliminate redundancy of the functional enrichment terms [64].

### 3.7. TUNEL

For apoptosis measurements, RNCs were fixed in 4% paraformaldehyde and sectioned, as previously reported [20]. Deoxynucleotidyl transferase-mediated dUTP nick end labeling (TUNEL) was performed to identify cells in the RNC explants that were undergoing apoptosis. FragEL DNA Fragmentation Detection Kit, Fluorescent—TdT Enzyme (Calbiochem, QIA39) was used and cells were quantified following the protocol previously adapted in our lab [20,65].

## 4. Conclusions

It is important to place our results within the context of the experimental procedure and comparison of tissues. For one, the stress elicited in our model system comes from two different procedures. On the one hand, it involves the transection of the RNC from anterior and posterior regions of the radial nerve, as well as the transection of peripheral nerves. On the other hand, the tissue was kept in a collagenase solution for ~24 h and then dissociated from the surrounding body wall. Therefore, the protocol provided a very particular environment, which may have impacted the expression of certain genes or signaling systems. This procedure can be compared with dissecting out portions of the spinal cord from the surrounding vertebra and cutting of the spinal nerves, then keeping them as explants in culture. Thus, some of the observed effects might be due to different “types” of stress, depending on the cells that may have suffered injuries or alterations in their neuronal circuits. Second, and most important, is that the tissues used for gene expression comparisons were RNCs kept in culture for three days following removal from the body wall by transection and enzymatic dissection. Comparisons to the in situ RNC were not possible, and it would be impossible to surgically isolate it without contamination from the surrounding tissues, as has been shown in some of our comparisons. Therefore, it is plausible that some of the observed responses were due to additional changes induced by the culture environment. Nonetheless, experimental data from the laboratory showed that RNCs kept in culture for up to 7 days maintain most of their morphological integrity and cell processes [20]. In addition, it is important to state that our comparative analyses implies that a gene that shows “no significant change” in its expression is either not changing at all in response to the dissection or, on the contrary, its expression changes take place after the enzymatic dissection and continues even after 3 days in culture.

In addition to these issues that pertain to our model, there are other issues associated with the transcriptomics/proteomics/genomic field that should be highlighted. First, is the fact that a change in mRNA levels does not necessarily imply a change in the expression of the protein. This is a controversial issue that depends on various factors, including the gene/protein itself, the cellular environment, and even the species of organism. Second, protein concentrations are not always correlated to changes in function, as proteins can be modified by post-translation changes that may go undetected depending on the methods used. Finally, differences in the expression of mRNAs and proteins should correlate with functional changes that reflect their biological relevance. In the end, only the full analyses of the interactions between molecules and their processes will allow us to unravel complex biological phenomena. Therefore, our data should be viewed as a first step in the long road of determining the molecular processes that allow echinoderms to undergo fully functional CNS regeneration.

Having stated these provisions, it still is important to highlight that the profile of gene expression attained in this study strongly suggests a stress response comparative to those detected in many tissues and animal species, but at the same time, with a gene expression profile that may be distinctive to the holothurian RNC. Most importantly, we provide evidence of stress-associated genes that are found within the holothurian nervous system transcriptome and show how some of these may play a role in the stress response (as determined by their changes in expression). Our work provides an important reference of genes (Table 5) that should be further studied to determine how the stress associated with injury might elicit some of the mechanisms found in the holothurians that protect their nervous system and allows full regeneration.

## Figures and Tables

**Figure 1 ijms-23-13393-f001:**
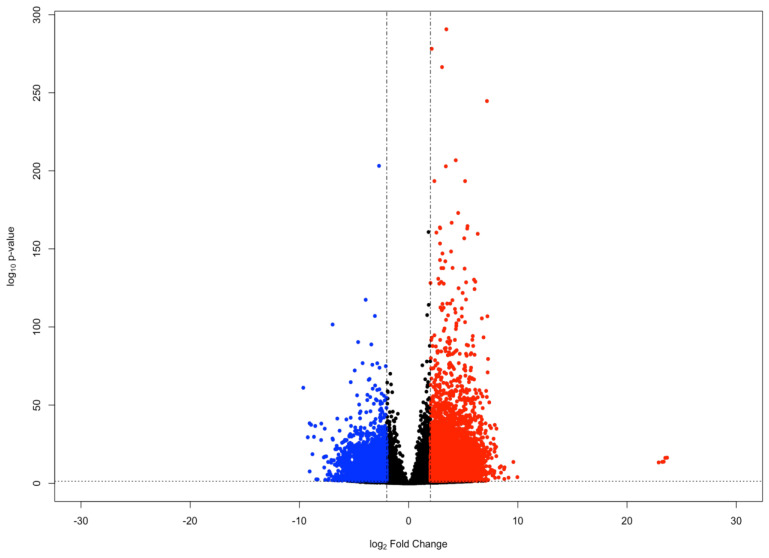
MA plot of genes in the holothurian CNS. Each point in the plot represents an individual gene. Points colored red represent upregulated genes (log_2_ Fold Change ≥ 2 and FDR-adjusted *p*-value < 0.05), whereas points colored blue represent downregulated genes (log_2_ Fold Change ≤ −2 and FDR-adjusted *p*-value < 0.05).

**Figure 2 ijms-23-13393-f002:**
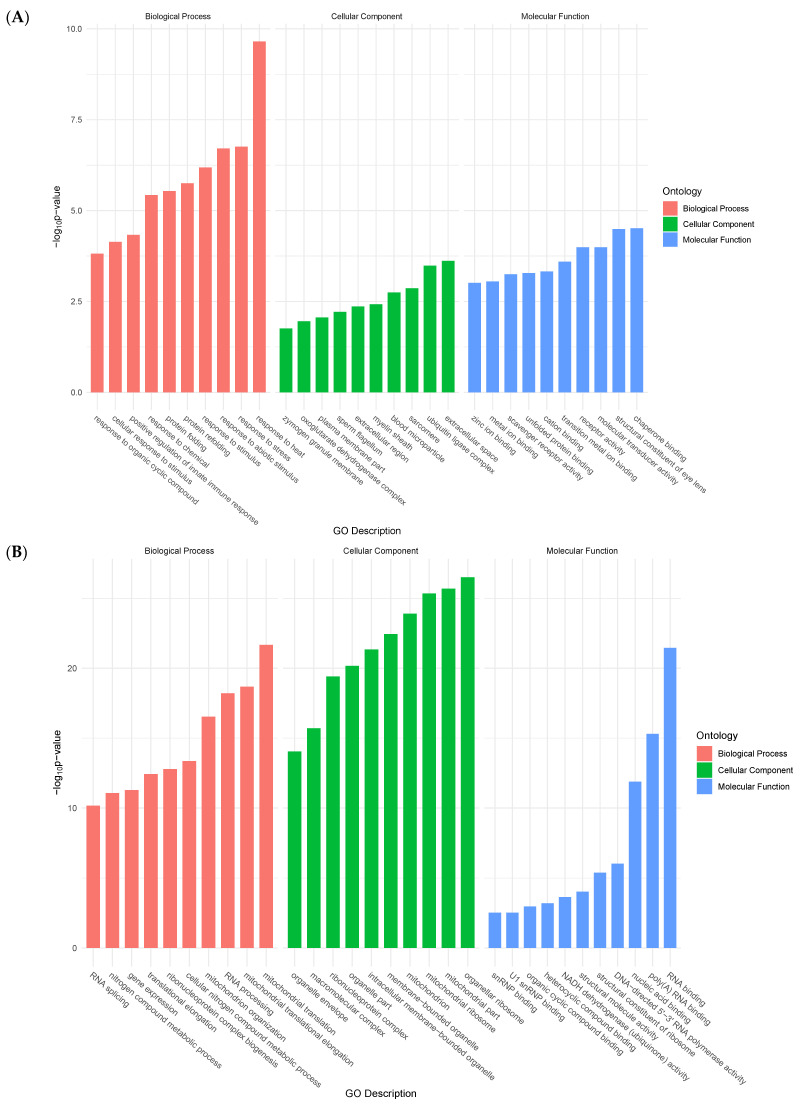
Bar plots showing top ten enriched functional terms for upregulated genes (**A**) and downregulated genes (**B**) at 24 h post-collagenase treatment.

**Figure 3 ijms-23-13393-f003:**
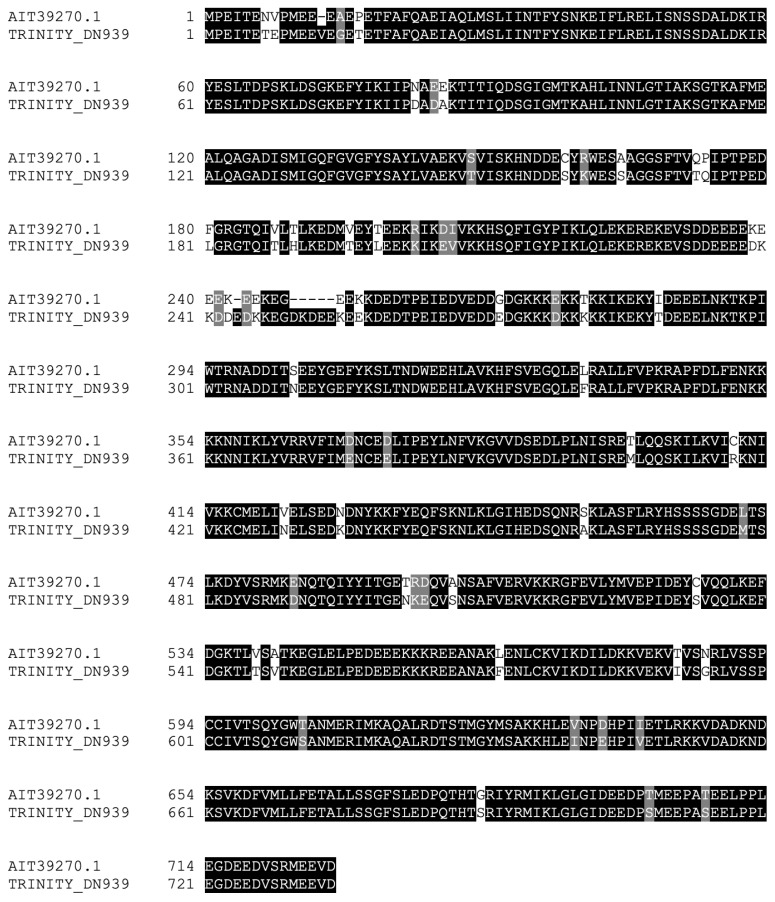
Alignment of *Hsp90* homolog identified in the RNC transcriptome with a *Hsp90* sequence from *A. japonicus*.

**Figure 4 ijms-23-13393-f004:**
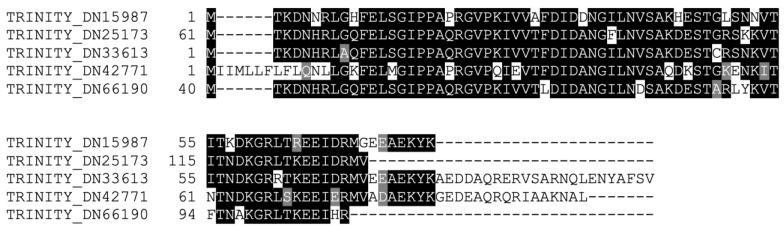
Alignment of the conserved domains of the *Hsp70* homologs identified in the RNC transcriptome.

**Figure 5 ijms-23-13393-f005:**
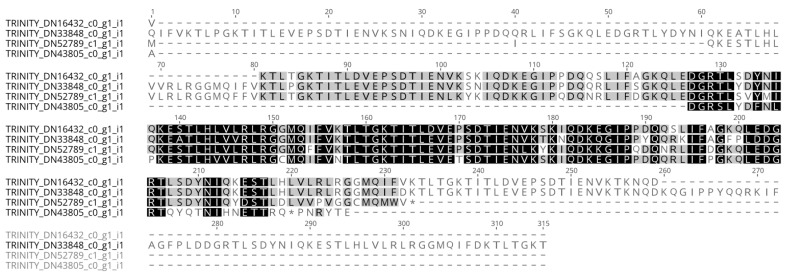
Alignment of the translated ORFs of the four distinct transcripts from the *H. glaberrima* RNC transcriptome that matched ubiquitin.

**Figure 6 ijms-23-13393-f006:**
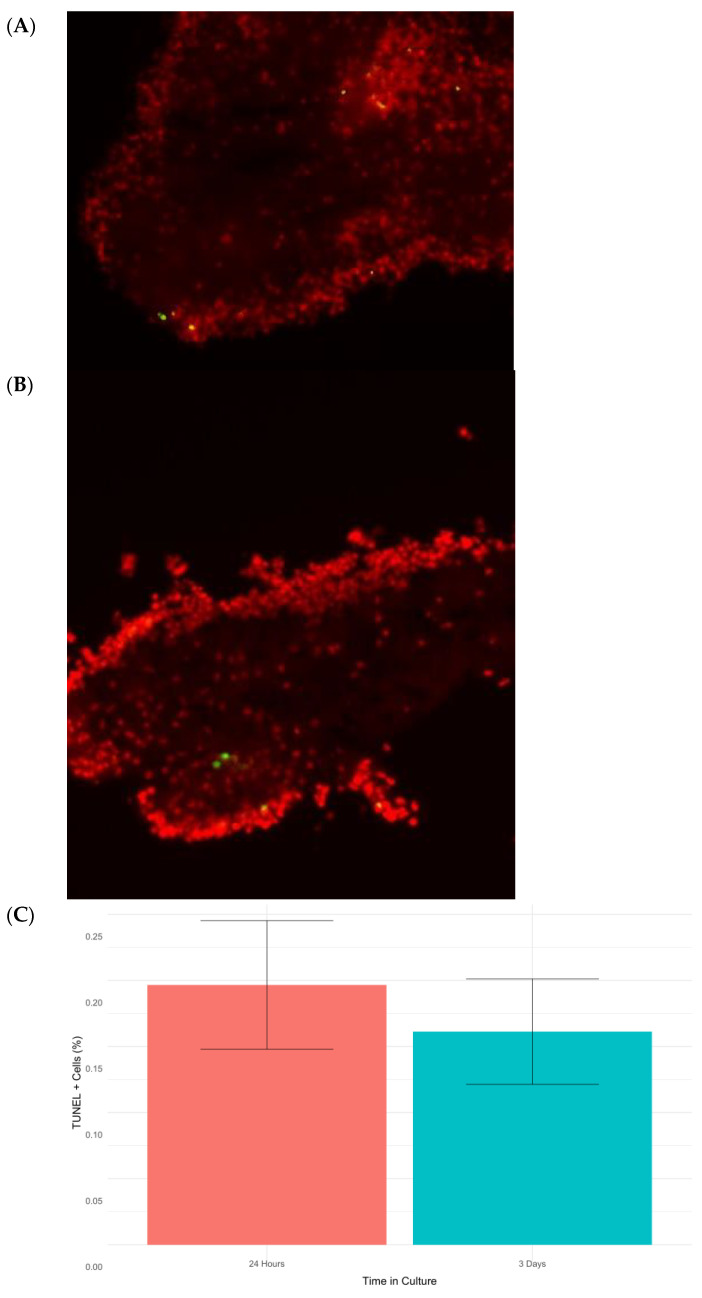
Micrographs of RNC explants at 3 days post-collagenase treatment (**A**) and 24 h post-collagenase treatment (**B**). Cell nuclei were stained with DAPI (red), and apoptotic cells were detected using TUNEL (green). Finally, the percentages of apoptotic cells in RNC explants placed in culture for 24 h and 3 d were calculated (**C**).

**Table 1 ijms-23-13393-t001:** Genes with most significant differential expression.

Gene	Uniprot Accession ID	Function	Pattern of Expression (vs. Day 3)
*RAB12*	Q6IQ22	Signal transduction	Downregulated
*THOC6*	Q86W42	Protein binding	Downregulated
*HSP40B*	Q3HS40	Unfolded protein binding	Upregulated
*TTLL7*	A4Q9F0	Cellular protein modification process	Upregulated
*POLYQ*	P23398	Protein binding	Upregulated
*RPS27A*	P62979	Structural constituent of ribosome	Downregulated
*HSPA1A*	P0DMV8	Damaged DNA binding	Upregulated
*TBC1D15*	Q8TC07	Rab GTPase activator activity	Upregulated
*EGR1*	A4II20	Differentiation and mitogenesis	Upregulated
*BCDO1*	Q9HAY6	Vitamin A metabolism	Upregulated
*NEURL1*	Q0MW30	E3 ubiquitin ligase	Upregulated
*HSPA2*	P54652	Unfolded protein binding	Upregulated
*JUN*	P54864	Transcription factor	Upregulated
*CBLIF*	P27352	Cobalamin binding	Downregulated
*RPS5*	P46782	Structural constituent of ribosome	Downregulated
*MRPL12*	Q7YR75	Structural constituent of ribosome	Downregulated
*SPDEF*	O95238	Transcription factor	Downregulated
*DMBT1*	Q9UGM3	DNA binding	Upregulated
*BAG3*	O95817	Heat shock protein binding	Upregulated
*DYRK2*	Q5ZIU3	Regulation of cell growth and development	Upregulated
*WARS2*	Q9UGM6	Mitochondrial tRNA ligase (tryptophan)	Downregulated
*SLC6A1*	P30531	GABA transporter	Upregulated
*TIMM8A1*	Q9WVA2	Mitochondrial intermembrane chaperone	Downregulated
*AP5S1*	Q9NUS5	Endosomal transport	Downregulated
*JRKL*	Q9Y4A0	DNA binding	Downregulated
*NETO2*	Q8NC67	Ionotropic glutamate receptor binding	Upregulated
*LZIC*	Q8WZA0	Beta-catenin binding	Downregulated
*NRL*	P54846	Transcriptional regulator	Upregulated
*FAM171B*	Q6P995	Integral component of membrane	Downregulated

Note: All genes have an adjusted *p*-value of <0.05.

**Table 2 ijms-23-13393-t002:** Apoptosis-related genes in the RNC transcriptome.

Gene	Gene Function	Pattern of Expression (vs. Day 3)
*TRAF2*	Apoptosis inhibitor	Upregulated
*AP-1*	Transcription factor involved in controlling proliferation, apoptosis, and proliferation	Upregulated
*SQSTM-1*	Promotes autophagy	Upregulated
*BAK1*	Promotes apoptosis	Upregulated
*CCAR-1*	Regulates cell division cycle and apoptosis	Upregulated
*HK1*	Carbohydrate metabolism and maintenance of outer mitochondrial membrane	Downregulated
*LAMTOR-5*	Inhibits apoptosis	Downregulated
*BIRC5*	Apoptosis inhibitor	No significant change
*XIAP*	Apoptosis inhibitor	No significant change
*BCL-2*	Apoptosis regulator	No significant change
*p53*	Apoptosis inhibitor	No significant change
*CASP1–3* and *6–8*	Apoptosis mediators	No significant change

Note: Differential expression significance threshold was set at adjusted *p*-value of <0.05.

**Table 3 ijms-23-13393-t003:** Integrated stress response factors.

Factors	Factor Name	Response	Pattern of Expression
*ATF4*	Activating Transcription Factor 4	Cell survival and recovery during ISR. ISR effector and master transcription factor	No significant change
*PERK*	PKR-Like ER Kinase	Kinases that initiate the ISR reponse by phosphorylating eIF2α translation initiation factor	No significant change
*PKR*	Double-stranded RNA-dependent Kinase		No significant change
*HR1*	Heme-regulated eIF2α Kinase		No significant change
*GCN2*	General control non-derepressible		No significant change
*IRE1*	Inositol-requiring protein 1	Initiate the UPR (Unfolded Protein Response)	No significant change
*ATF6*	Activating transcription factor 6		No significant change
*PP1*	Protein Phosphatase 1	ISR termination signal	No significant change
*eIF2a*	Eukaryotic translation initiation factor 2A	Phosphorylation of this factor commences the ISR	No significant change
*CHOP*	C/EBP homologous Protein	CHOP/ATF4 interaction negatively regulates ATF4	Not found

**Table 4 ijms-23-13393-t004:** Transcription factors with significant differential expression.

Gene	Pattern of Expression (vs. Day 3)
*ATF2*	Upregulated
*RXRA*	Upregulated
*EF1A1*	Upregulated
*ZNFX-1*	Upregulated
*NR4A2*	Upregulated

Note: All genes have an adjusted *p*-value of <0.05.

**Table 5 ijms-23-13393-t005:** All genes mentioned in the body of this text and their respective changes in expression.

Gene	Log_2_ Fold Change	FDR-Adjusted *p*-Value
*RAB12*	−3.52	0.0006
*THOC6*	−3.91	9.90 × 10^−6^
*HSP40B*	6.67	1.75 × 10^−6^
*TTLL7*	5.96	0.0002
*POLYQ*	3.24	0.0001
*RPS27A*	−4.71	0.0009
*HSPA1A*	5.48	3.32 × 10^−8^
*TBC1D15*	5.40	1.25 × 10^−5^
*EGR1*	6.75	3.62 × 10^−25^
*BCDO1*	5.93	1.13 × 10^−16^
*NEURL1*	4.79	3.77 × 10^−14^
*HSPA2*	5.97	5.93 × 10^−13^
*JUN*	4.55	1.31 × 10^−5^
*CBLIF*	−4.76	1.66 × 10^−11^
*RPS5*	−8.43	6.42 × 10^−9^
*MRPL12*	−5.81	1.98 × 10^−8^
*SPDEF*	−5.38	1.09 × 10^−7^
*DMBT1*	5.93	0.0008
*BAG3*	4.77	6.05 × 10^−7^
*DYRK2*	3.43	6.25 × 10^−7^
*WARS2*	−4.43	0.0001
*SLC6A1*	4.33	7.71 × 10^−7^
*TIMM8A1*	−5.78	9.03 × 10^−7^
*AP5S1*	−4.67	1.22 × 10^−6^
*JRKL*	−7.14	2.68 × 10^−6^
*NETO2*	4.27	2.77 × 10^−6^
*LZIC*	−4.23	3.12 × 10^−6^
*NRL*	5.86	4.29 × 10^−6^
*FAM171B*	−3.73	5.77 × 10^−6^
*HSP20*	3.30	0.0002
*HSP90*	4.77	1.20 × 10^−7^
*HSP70_1*	4.39	1.24 × 10^−5^
*HSP70_2*	6.73	2.30 × 10^−8^
*HSP70_3*	5.17	1.07 × 10^−7^
*HSP70_4*	3.52	0.0002
*HSP70_5*	5.54	9.24 × 10^−7^
*DNAJC29*	3.34	0.00023117
*CALR*	−5.59	5.90 × 10^−6^
*BCS1*	−3.51	0.00016118
*TIMM10*	−4.97	3.85 × 10^−5^
*RBBP6*	2.49	0.0007
*RNF31*	2.54	0.0004
*UBE3C-E3*	3.07	0.0008
*TRIM2*	3.58	0.0008
*β-TrCP*	4.43	0.0003
*DYRK2*	3.43	6.25 × 10^−7^
*ZFAND2B*	4.11	3.48 × 10^−5^
*NOP53*	−3.98	0.0008
*UBLCP1*	−2.53	0.0006
*BABAM1*	−3.99	0.0003
*UBFD1*	−4.00	0.0005
*UBE2C*	−4.45	0.0009
*STUB1*	−3.11	0.0009
*CUL5*	−3.28	0.0005
*UBE2U*	−4.39	4.29 × 10^−5^
*UBB_1*	3.33	1.09 × 10^−5^
*UBB_2*	3.15	1.25 × 10^−5^
*UBB_3*	3.20	0.00029068
*UBB_4*	3.87	1.20 × 10^−5^
*TRAF3*	2.38	0.0006
*AP-1*	4.41	6.46 × 10^−7^
*SQSTM-1*	2.47	0.0002
*BAK1*	2.88	0.0002
*CCAR1*	5.64	0.0008
*HK1*	−3.31	0.0004
*LAMTOR-5*	−4.58	0.0001
*XBP-1*	2.73	0.03
*GYPSY-19*	3.66	0.0001
*GYPSY-1*	−3.89	0.0002
*HERO7*	−2.62	0.001
*ATF2*	3.25	0.0006
*RXRA*	4.81	0.0009
*EF1A1*	4.41	6.91 × 10^−5^
*ZNFX-1*	6.53	3.97 × 10^−6^
*NR4A2*	2.91	0.0001

## Data Availability

All sequencing data for this study are publicly available in our laboratory website (https://blastkit.hpcf.upr.edu/hglaberrima-v1/). A table with differential expression data and annotations for each contig is also available at https://devneurolab-data.shinyapps.io/dge-web/ (all accessed on 2 July 2020).

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
