# Peer review of "The Stress Response of the Holothurian Central Nervous System: A Transcriptomic Analysis"

_ijms, 2022, doi:10.3390/ijms232113393_

Round 1
Reviewer 1 Report
This study addresses an important question, i.e., the factors that mediate successful or un-successful regeneration in nervous tissue. The approach taken is to study the expression of various genes in the radial nerve cords of a species of sea cucumber that have been subjected to collagenase treatment. Comparison is made between acutely treated nerve cords and others that have been left in culture for 3 days. The choice of genes to investigate is based largely on prior literature in which various candidates have emerged as playing roles in successful or unsuccessful regeneration. As might be expected, many genes exhibited either up- or down-regulation when comparing the two conditions. The authors then speculate on how these changes relate to other reports on the roles of various factors in either promoting or inhibiting successful regeneration with a special focus on apoptosis.
The work appears to have been carried out with appropriate use of the methods and the manuscript is written in clear English with proper use of grammar and vocabulary. The authors clearly state some of the limitations of their study, including the difficulty of determining the appropriate control with which to compare the changes in gene expression. On this latter point, I wonder if it might be possible to see whether there are comparable changes in gene expression when subjecting the 3-day culture explants to additional injury or collagenase treatment? The authors make a good case for the idea that some of the prior changes that have been reported may be due to the inclusion of non-neural tissue. However, it is also possible that the surrounding tissues make their own contribution to the response to injury.
The latter point raises what to me is the most significant limitation of this study, which is the reliance on changes in gene expression to infer biological relevance. Although such reductionistic approaches have become commonplace and widely used, I remain skeptical that this provides the kind of information that is ultimately going to be useful in unraveling complex biological phenomena. Even the notion that genes can be mapped onto specific biological functions, e.g., Table 2, is controversial. Gene functions are highly dependent on the context in which they are expressed. The emphasis given to regulators of apoptosis is intriguing but given the very low percentage of apoptotic cells in this model, it is not clear what the next step would be to pursue this line of investigation. It is possible that the data provided in this paper will eventually give rise to specific hypotheses regarding the differences between nerve tissue that successfully regenerates and that which does not. At this stage, it is not obvious that cataloguing such changes will guide future research without attention to higher-level tissue functions.
Reviewer 2 Report
Comments:
In this article, author has explored the gene activation profile of the stressed holothurian CNS which has regenerative capacity to stress response and inflammation by the injury. The results provide a profile of dysregulated gene and pathways which undergo CNS regeneration.
Major comments:
· Results are based on only RNAseq analyses performed on the RNC data. No experiments are performed on proteomics to validate the gene regulation.
· Author should plan some experiments to study the proteomics of highly dysregulated proteins either by western blot analysis or quantitative mass spectrometry.
· Conclusion part must be improved.
Round 2
Reviewer 2 Report
NA